# Single drop cytometry onboard the International Space Station

Daniel J. Rea [1,2,9], Rachael S. Miller [3,9], Brian E. Crucian[4], Russell W. Valentine[5], Samantha Cristoforetti[6], Samuel B. Bearg[1,2], Zlatko Sipic[1,2], Jamie Cheng[1,2], Rebecca Yu[1,2], Kimesha M. Calaway[5], Dexter Eames[7], Emily S. Nelson[8], Beth E. Lewandowski[8], Gail P. Perusek[8] & Eugene Y. Chan [1,2] ✉

Real-time lab analysis is needed to support clinical decision making and research on human missions to the Moon and Mars. Powerful laboratory instruments, such as flow cytometers, are generally too cumbersome for spaceflight. Here, we show that scant test samples can be measured in microgravity, by a trained astronaut, using a miniature cytometry-based analyzer, the rHEALTH ONE, modified specifically for spaceflight. The base device addresses critical spaceflight requirements including minimal resource utilization and alignment-free optics for surviving rocket launch. To fully enable reduced gravity operation onboard the space station, we incorporated bubble-free fluidics, electromagnetic shielding, and gravity-independent sample introduction. We show microvolume flow cytometry from 10 μL sample drops, with data from five simultaneous channels using 10 μs bin intervals during each sample run, yielding an average of 72 million raw data points in approximately 2 min. We demonstrate the device measures each test sample repeatably, including correct identification of a sample that degraded in transit to the International Space Station. This approach can be utilized to further our understanding of spaceflight biology and provide immediate, actionable diagnostic information for management of astronaut health without the need for Earth-dependent analysis.

Under National Aeronautics and Space Administration (NASA) plans, humans are to travel to Mars in the 2030s[1]. For humans to explore Mars, the journey is a 1.8-billion-kilometer round-trip journey, requiring a total of 760–850 days[2]. This is greater than the 1000× more distance traveled and close to the 100× more duration than Apollo 11, which was 1.534-million-kilometers round-trip and just over 8 days[3]. The human health risks are significant, including space radiation exposure[4], bone loss[5], circadian rhythm changes[6], spaceflight psychological hazards[7], cardiac remodeling[8], vision changes[9], hematological dysfunction[10], and neurological changes[11]. Unexpected, acute, life-

threatening medical conditions can arise that require emergent diagnostic assessment and medical intervention, such as the case of an obstructive jugular venous thrombosis on the International Space Station (ISS)[12]. The risk of inflight medical conditions is among the most concerning for a Mars mission per NASA's Human System Risk Board[13]. For humans to become an interplanetary species, these risks need to be studied and managed. In-flight clinical decision-making can benefit from immediate and abundant diagnostic information, available from drops of blood or other biological specimens that can be easily and frequently obtained. Separately, the analysis of biological samples is a

[1]DNA Medicine Institute (DMI), Bedford, MA, USA. [2]rHEALTH, Bedford, MA, USA. [3]KBR, Houston, TX, USA. [4]Human Health and Performance Directorate, NASA Johnson Space Center, Houston, TX, USA. [5]ZIN Technologies, Middleburg Heights, OH, USA. [6]European Space Agency, Paris, France. [7]Graylark, Cambridge, MA, USA. [8]NASA Glenn Research Center, Cleveland, OH, USA. [9]These authors contributed equally: Daniel J. Rea, Rachael S. Miller. ✉e-mail: echan@dnamedinstitute.com

component of spaceflight research, but currently, the samples are downmassed, which can result in sample transportation artifacts. This process takes months and during this time, the samples can be subject to unpredictable storage conditions, resulting in degradation or alteration. This was highlighted in the monozygotic twin study where samples had unavoidable transit time and unknown transit conditions (vibration and temperature)[14]. Our current understanding of space-flight medicine and biology is thus limited by this approach, and while sample downmass, however imperfect, is an option for studies in Low-Earth Orbit (LEO), it would be near impossible for long-duration, deep space exploration missions.

There has been a longstanding interest in developing and imple-menting a cytometer for routine spaceflight use. This would allow for immediate sample analysis, without the risk of sample transport arti-facts or delays in results. Flow cytometry capabilities in space have been desired for decades. Jett et al., in 1985, described the ability to leverage cytometry on the space station, a lunar base, or a voyage to Mars[15]. This approach is powerful enough to measure a broad range of test classes, such as blood counts[16], hormones[17], chemistry[18], enzymes[19], nucleic acids[20,21], proteins, and biomarkers[15,22]. Further-more, cytometry can allow for high levels of assay multiplexing, allowing simultaneous measurement of diagnostic and biological parameters, thus increasing the throughput and content of each sample analysis. For instance, multiplexing over a 100 analytes is possible with differentially dyed microspheres[23] and multiplexing in the 1000s is possible with barcoded hydrogel microparticles and nanostrips[24–26]. Thus, cytometry with multiplexing capabilities would provide the required data density for detailed insights into biological systems and astronaut health. This breadth and depth of applications make a cytometer highly desirable for spaceflight applications.

There are multiple challenges, however, to implementing a cyt-ometer in space for routine use[27,28]. The first is obvious, which is the mass, volume, and power constraints. For instance, a BD LSRFortessa X-20 is large at 159 kg, 76.2 × 73.7 × 76.2 cm, and 1500 W. For context, this would exceed the mass and volume allocation of the entire spacecraft medical system. A smaller cytometer, such as the BD Accuri C6, is 13.6 kg, 27.9 × 37.5 × 41.9 cm, and 150 W. Even this smaller, cap-able cytometer would consume significant resources. In addition to the resource issues, cytometers require sensitive laser alignment, generally focused down to the sample stream core, which is about 1/10th the width of a 200 µm diameter human hair. The ability of this delicate system to survive a rocket launch, with high *g*-loads and vibration, would be very challenging. Even if transported to space safely, the cytometer would need to be able to operate properly in microgravity. The fluidics in the system would be prone to air bubbles since there is no buoyancy in space. Air bubbles would occupy the middle of the liquids and be likely to interfere with the sheath flow operation. This can manifest in sheath flow stream drift relative to the laser, degrading the performance of the system. Cytometers generally also require a significant level of calibration and routine weekly maintenance to keep them running so optimal data can be obtained. Any payload going to space is likely to be shipped to the launch facility months in advance, precluding any servicing during this time. Training the astronaut crew members to service these complex instruments and the performance of the calibration procedures would be significantly time-consuming.

Several groups have made advancements in demonstrating cytometry in space or a space analog environment. The MicroFlow1 was demonstrated onboard the ISS by the Canadian Space Agency (CSA)[29]. It achieved suitable performance but lacked a fluidic system for loading small samples in microgravity. The authors from DNA Medicine Institute demonstrated a miniaturized solid-state flow cyt-ometer (an early version of the rHEALTH ONE), onboard parabolic flights for cell and nanoscale test strip (nanostrip) measurements[25,26,30], together with a microvolume in-line capillary sample loader[31–33] and a

microfluidic spiral vortexer for mixing and dilution[34]. Others have also shown promise in addressing the challenges of cytometry in reduced gravity environments, including a Guava cytometer significantly modified for microgravity operations[35], a 460 nm blue LED-based cytometer[36], and a plastic chip-based fiber optic cytometer[37]. All these technologies, including the rHEALTH ONE precursor, leveraged a sheathless approach. Sheath flow under hydrodynamic focusing, however, is the standard approach on larger cytometers. This approach brings the sample off the walls of the channel for less channel fouling and avoids the zero-flow condition at the flow chan-nel's walls. Sheath flow further allows the sample core stream to be centered in the channel, where the Poiseuille flow rate is most uniform.

Cytometers are generally designed for environments with a gravity vector and a stable work surface, preventing their off-the-shelf use in space. Gravity assists with minimizing air bubbles in fluidic systems by relying on buoyancy. In 1*g*, fluids are in predictable loca-tions, at the bottom of a vial or a vessel. In microgravity, the locations of fluids can be unpredictable and dictated by simple movement and/or surface forces. This necessitates the development of microgravity methods for handling cytometry fluids and sample introduction. These approaches should prevent or minimize microbubble formation, which degrades cytometry data. On the optical side, commercial cyt-ometers generally have optomechanical positioners that allow tech-nicians to fine-tune the performance and alignment of the laser(s) relative to the flow cell and detectors. Since the positioning tolerance is less than the width of a human hair, systems need to be realigned routinely. Some cytometers employ a fixed alignment approach, but these would not be rated to the *g*'s and vibrations experienced on rocket launch. Rocket launch conditions may also result in loose electrical, optical, or mechanical conditions, resulting in catastrophic failure of equipment. Furthermore, critical spacecraft communications may be impacted by electromagnetic interference (EMI) coming from high-powered microprocessors required for data collection and ana-lysis. These reasons as well as significant resource limitations for mass, volume, power, and fluids ultimately preclude launching a commercial cytometer to space and having it yield useful information.

Here, we describe the analysis of individual drops of test sus-pensions with a spaceflight-modified rHEALTH ONE, a sheath-flow, cytometry-based biomedical analyzer[38], onboard the ISS. The device was designed and built with spaceflight considerations by the authors from DMI and rHEALTH, leveraging on previous successes with reduced gravity testing onboard parabolic flights[30]. The base rHEALTH ONE addresses the need for minimal mass, volume, and power (1.5 kg, 12 × 13 × 18 cm, and 2.9 W), alignment-free optics, and single drop sample handling. This base device considered a Commercial-Off-The-Shelf (COTS) device by NASA, was further developed as a payload for the space station, which included addressing the conditions of rocket launch, microgravity sample loading techniques, microgravity fluid bottles, safety, and EMI. The resulting device was fully operational in space. It leveraged sheath flow for precise measurements of test microspheres that were designed to comprehensively assess system performance. Sample loading from individual sample drops was achieved with a zero dead volume sample loading system for the analysis of precious samples. Preflight, in-flight, and post-flight analysis verified the robust operation of the device on-orbit.

## Results

### rHEALTH ONE experiment and device description

The rHEALTH ONE is a portable, microvolume sampling, dual-laser, and five-channel cytometer employing hydrodynamic focusing (spe-cifications, Supplementary Table 1). The rHEALTH ONE was sent to the ISS as part of the NASA Commercial Resupply Mission NG-17. An Antares rocket carried the NG-17 Cygnus spacecraft to the ISS, after launching off the Mid-Atlantic Regional Spaceport (MARS) Pad 0A on Wallops Island, Virginia on February 19, 2022. The details of the pre-

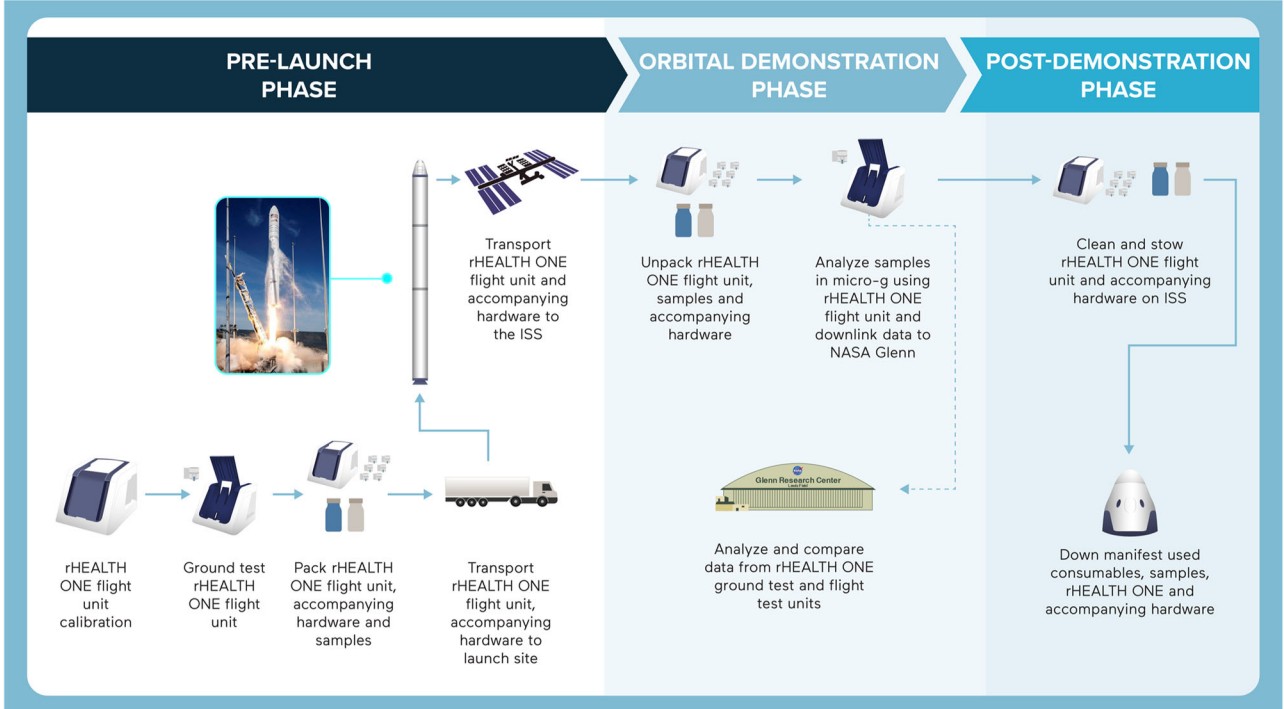

**Fig. 1 | Summary of the logistics of the rHEALTH ONE on-orbit demonstration.** Left: the pre-launch phase included calibration of the unit, ground testing, packing, transportation to the launch site, launch, and ISS docking. Antares rocket image credit: Northop Grumman. Middle: the orbital demonstration phase included unpacking, analysis of the samples in microgravity with the rHEALTH ONE, and data downlink to NASA Glenn Research Center. Right: the post-demonstration phase included cleaning and stowage of the unit, downmass of test hardware and samples, and adjudication of discrepant samples.

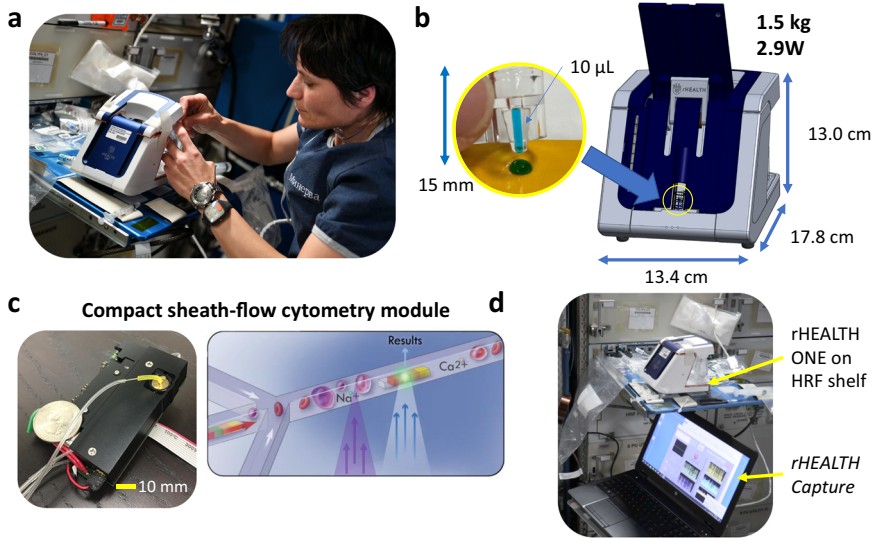

**Fig. 2 | Overview of the rHEALTH ONE experiment on the ISS. a** The rHEALTH ONE during device setup by ESA astronaut Samantha Cristoforetti. Image courtesy of NASA. **b** Sample drop shown being wicked up using the capillary-based sample consumable, which is then loaded into the sample loader, located in the front of the device. The device is 1.5 kg with dimensions of 13.4 × 17.8 × 13.0 cm. **c** A compact, sheath-flow cytometry module, with a US quarter for size reference, performs the measurements. Sheath-flow hydrodynamic focusing aligns cells and particles for one-by-one analysis through a 405 and a 532 nm laser. **d** The on-orbit runs were performed on a HRF shelf and data were streamed to an ISS laptop running the *rHEALTH Capture* software program.

launch, orbital demonstration, and post-demonstration phases are outlined in Fig. 1. SpaceX Crew-4 Commander and European Space Agency (ESA) astronaut Samantha Cristoforetti unstowed the system contents (Supplementary Fig. 1) and performed experiments to characterize the device on May 13, 2022 between 09:00 and 17:00 GMT and again on May 16, 2022 between 10:45 and 18:30 GMT[39,40].

The device was mounted on a Human Research Facility (HRF) shelf (Fig. 2a). Four samples were flown to test the device's optical alignment, precision, intensity resolution, size resolution, and spectral separation (see sample details in Supplementary Table 2). These samples were blinded to the authors at DNA Medicine Institute and rHEALTH until the day of on-orbit operations. The samples were designed to be safe for

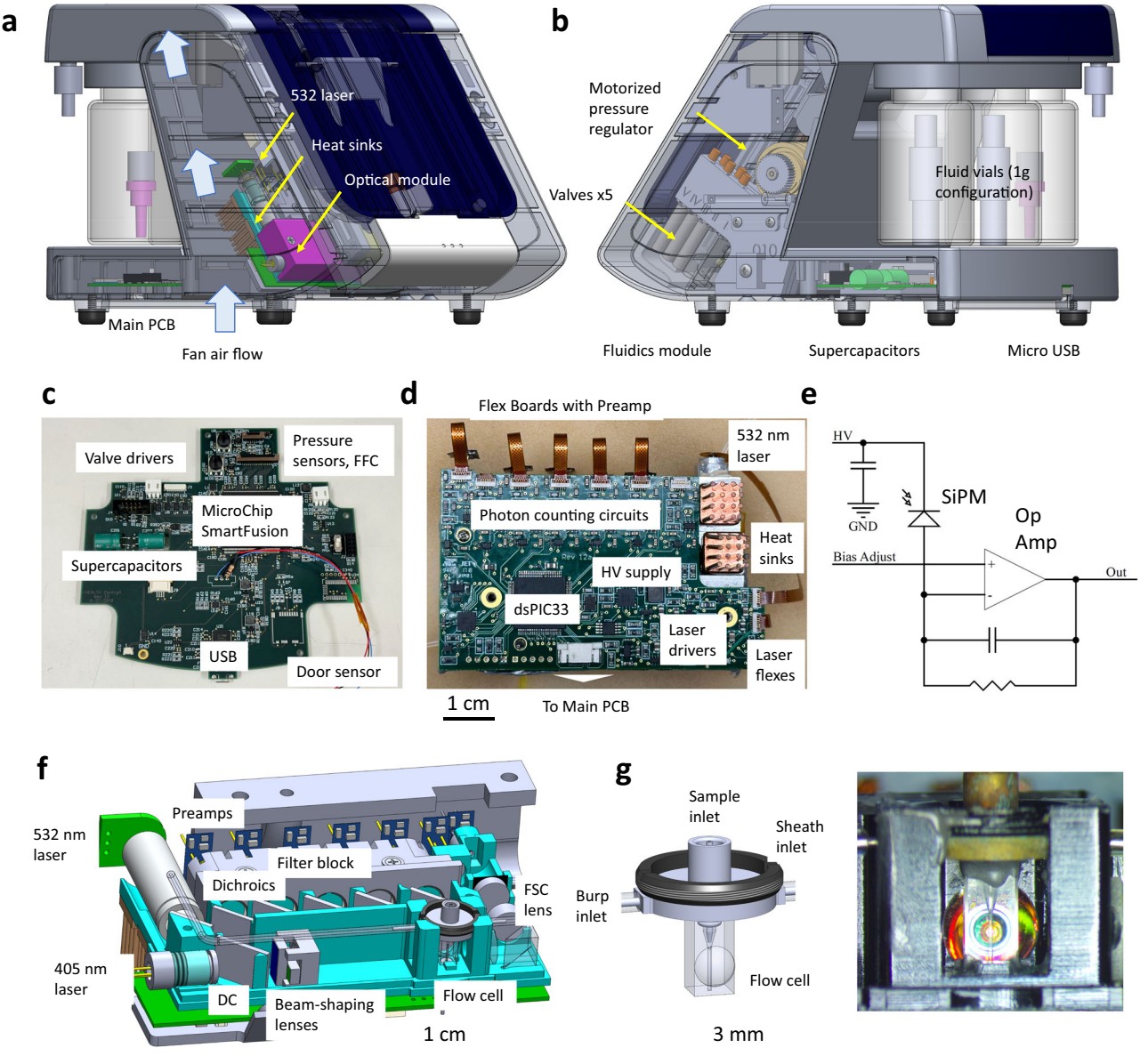

**Fig. 3 | rHEALTH base device with detail of subsystems. a** Transparent image of the device, showing the location of the optical module in the unit. The 532 nm laser and associated copper heat sinks are directly in the path of the fan. **b** The fluidics module is located on the opposite side of the optical module. It is a microfluidic assembly with a bank of five low-power latching valves and a motorized pressure regulator. **c** Main PCB that controls the device and sends data to the USB-attached PCB. The valve drivers are on the side of the fluidics module. Supercapacitors allow for additional power for valve switching. A field-programmable gate array (FPGA)-based system on a chip (SmartFusion) provides computing. Wires lead to a door sensor. A flat flex connector (FFC) provides data and power between the main PCB and the detector PCB. **d** The back of the optical module has an integrated detector PCB with attached flex boards with preamplifiers, powered by the high-voltage (HV) supply. The detector PCB has a separate microprocessor (dsPIC33) to provide counters for photon counting. Additional features are labeled in the figure. **e** Preamplifier circuit schematic. Photons captured by the SiPM result in a detectable signal after the op-amp. HV, ground (GND) provides power input and the photon counting signal is the Out. **f** Inside the optical module showing the locations of the 532 and 405 nm lasers, bandpass filters, dichroic filters (DC), lenses, and overall layout. **g** Left: detail of the flow cell showing the retaining ring, flow cell top (with burp, sample, and sheath inlets), and fused silica flow cell with integrated lens. Right: fused silica flow cell image with epoxied brass flow cell top as in the graphic.

use in the cabin without the need for additional levels of containment. Sample drops were dispensed from dropper bottles onto polyimide tape, selected for their ability to form a beaded drop. Capillary action was utilized to load the rHEALTH ONE sample consumable (Fig. 2b). Microgravity assisted the filling of the sample consumable as capillary forces did not have to compete with the sample's hydrostatic pressure, as would be the case in 1*g*. Using this approach, the astronaut operator was able to consistently fill the 10 µL volume of the sample consumable each time. This wicking approach was similar to that used for testing the Hemocue WBC DIFF cuvette on-orbit[41]. The sample was loaded into the in-line sample loader of the device, which allows the entire sample to be analyzed. This approach is in contrast to conventional flow cytometers that require more volume than is analyzed, which results in a significant unanalyzed dead volume. Once loaded within the analyzer, the sample is delivered via pressure-driven, sheath-flow-based hydrodynamic focusing to the laser illumination region of the cytometry module (Fig. 2c, Supplementary Fig. 2). To meet the minimal resource requirements, the optical cytometry module occupies a volume less than $80 \times 50 \times 10$ mm, requiring no more than 1 W power. This palm-sized module has all solid-state components including a 405 nm laser, a 532

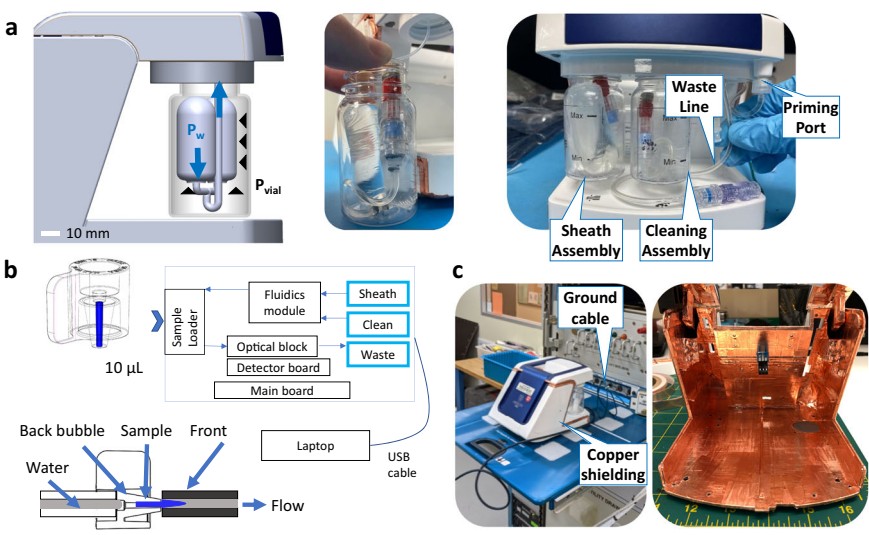

**Fig. 4 | Spaceflight modifications required for on-orbit operation. a** Pressurized clear plastic bottles provide the driving force for filtered water inside flexible bags. Left: the air space between the bottle and the flexible bags is pressurized ($P_{vial}$) and fluid pressure inside the bag ($P_{water}$) drives the sample into the cytometry module. Middle: filled fluid bag with the connectors prior to assembly into the back of the unit. Right: The back of the device shows the sheath and cleaning bottle assemblies (each with filtered water), a waste line prior to connection to the waste bag, and a priming port. **b** Schematic of the sample consumable, sample loading, and instrument block diagram. The sample consumable containing 10 µL sample (blue) is loaded into the sample loader that generates a fluid–fluid interface at the leading edge and an intentional air bubble behind the sample. The sheath bottle holds water that drives the sample into the optical block and to the waste. The water in the cleaning bottle rinses the system between runs. **c** The device (shown on a ground HRF shelf) had an additional grounding cable attached. Electromagnetic shielding (copper tape) is installed on the inside of the device.

Nd:YAG laser, and five silicon photomultipliers (SiPM). Sheath-flow encases the sample, bringing the sample off the walls of the detection channel, into the channel center for uniform flow. The sample passes through the 405 nm laser spot first, followed by the second 532 nm laser spot. During the runs, data are streamed via the USB cable to an ISS laptop running the *rHEALTH Capture* software program, which reports the number of photons collected by the five detectors in 10 µs intervals. Each sample run was approximately 2 min, yielding an average of 71.55 million raw data points for each run. The data were streamed in real-time, allowing visualization of each run by the astronaut (Fig. 2d).

In order to successfully perform cytometry onboard the ISS, unique engineering requirements had to be met. The base device, developed with NASA high-level requirements but prior to the payload development process, is shown in Fig. 3. The optical module and fluidics module are mounted inside the unit on both sides (Fig. 3a, b). A cooling fan on heat sinks is mounted to the optical module. Since there is no natural convective flow in microgravity, the fan is a must for proper device operation. The fluidics module has valving and a pressure regulator for controlling the sample, sheath, and cleaning fluids[42]. Custom printed circuit boards (PCBs) include the main board and the detector board with parallel SiPM circuits (Fig. 3c–e). The optical module has fixed alignment with all components epoxied in place. No adjustable positioners were used in the design (Fig. 3f)[43]. The flow cell is fabricated from low-autofluorescence fused silica with an integrated half-ball lens (Fig. 3g). A brass flow cell top with press-fit hypodermic gauge tubes allows for sheath, burp, and sample connections. This flow cell is integrated into the optical module to allow precise positioning relative to the lasers. The result is a plug-and-play cytometry module that is readily integrated with the rest of the system. Unlike conventional cytometer optics and flow cells, the result is an alignment-free module that maintains relative positioning, at the micron scale, between the lasers, the flow cell, and the detectors.

## Spaceflight ruggedization and modifications

Specific spaceflight modifications were required as part of the payload development process. The fluidic system had to operate without buoyancy, have minimal air bubbles, and accommodate a 10 µL sample. A pressurized fluidic system specifically designed for microgravity was implemented by using a flexible fluid-filled bag within the sheath and cleaning bottles in the back of the instrument. Using a syringe, the astronaut filled the bags with filtered water and utilized a figure-eight swinging maneuver to remove any air bubbles. The bag was a 0.014 mm thick, easily deformable medical-grade balloon, allowing the system to operate at its intended pressure of 70 mbar. Pressure external ($P_{vial}$) to the fluid bag provides the driving force ($P_w$) into the device (Fig. 4a). Other microgravity considerations included using a disposable waste bag with a unidirectional check valve to contain the test fluids after the runs. This replaced the standard, gravity-based waste bottle. Crew instructions provided key details on achieving bubble-free fluid bags and samples (Supplementary Fig. 3). The sample wicking procedure developed for spaceflight required testing with the sample loader. The loader has an in-line mechanism that forms a seal around both ends of the sample consumable, which has a 10 µL internal capillary volume (Fig. 4b). It generates a defined sample loading fluid profile with a properly wicked sample, marked with Poiseuille flow in the sample's leading edge and a fluid bubble in the back of the sample to allow the entire sample to be delivered to the cytometry module for absolute volumetric particle counts. This specific fluid loading profile is critically important for full analysis of the sample when compared with other loading profiles (Supplementary Fig. 4). To meet the ISS electrical requirements, copper tape was applied to the inside walls of the device for electrical shielding and an additional grounding cable was added (Fig. 4c). Finally, the system was fully ruggedized to withstand the high vibration and *g*-loads experienced on launches and returns (see Supplementary Tables 3, 4 and Supplementary Fig. S5). All fluidic joints were reinforced with either zip ties or waterproof heat shrink tubing. Electrical connections were reinforced

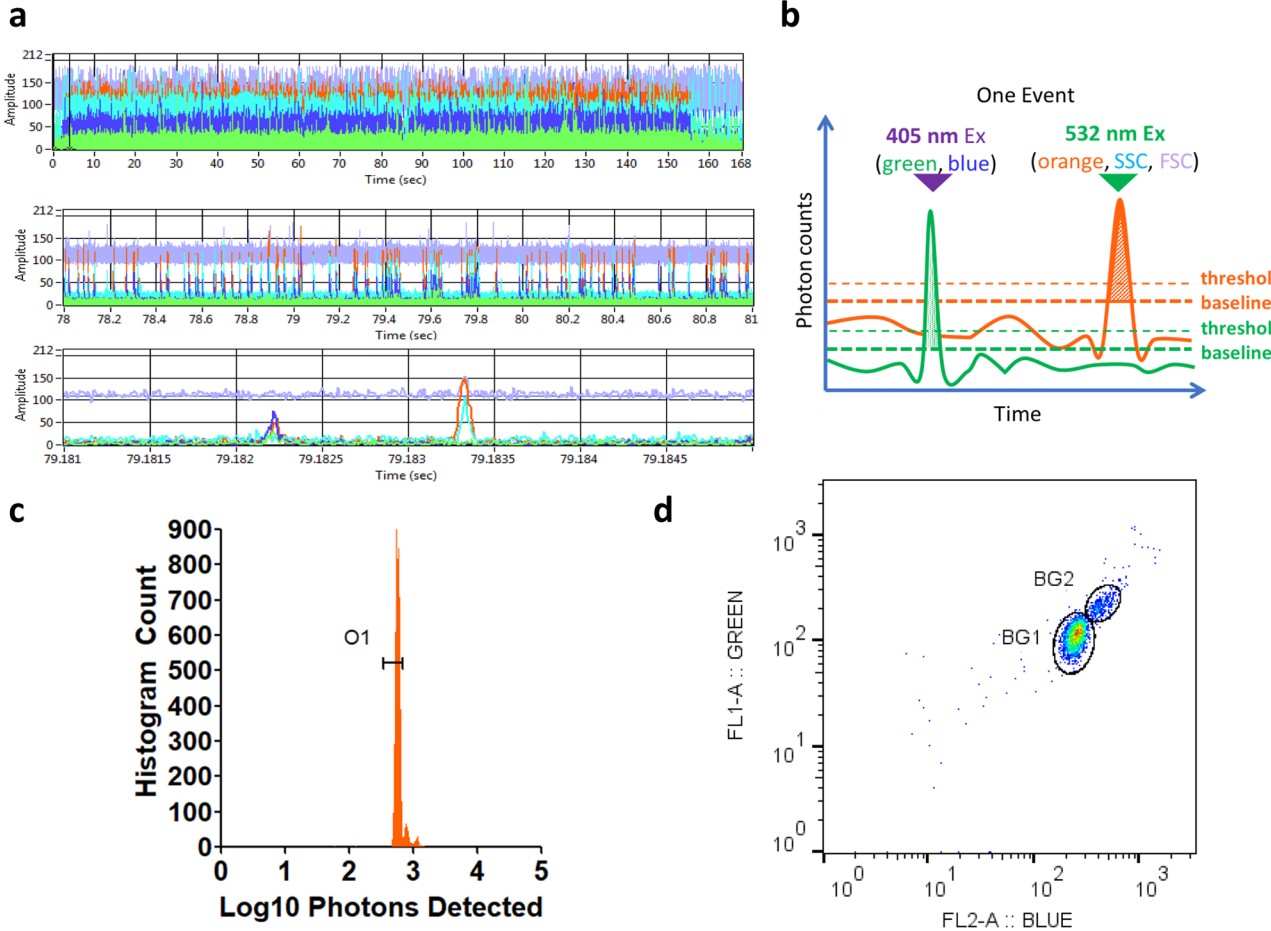

**Fig. 5 | Data output for a Flow-Set microsphere run. a** Raw data from five channels, collected in 10 µs intervals, visualized in the *rHEALTH Viewer*. The sample is Flow-Set microspheres. The top shows the full run, which lasts for 155 s prior to the appearance of the trailing edge air bubbles. The channel colors are as follows: purple (FSC), cyan (SSC), dark blue (blue channel), green (green channel), and orange (orange channel). The middle shows a zoom-in to 3 s of data, allowing individual events to be seen. The bottom is a 4 ms window showing a single event. The first peak is the transit of the 3 µm microsphere through the 405 nm laser (blue and green channels) and the second is through the 532 nm laser (FSC, SSC, and orange channels). **b** The software analyzes burst intensity for each peak after determining a suitable threshold (dashed lines) and baseline (bold dashed lines). For illustrative purposes, only the green and orange channels are shown. The burst intensity is integrated into the baseline. **c** The orange burst intensities are plotted on a histogram showing the number of counts (*y*-axis) at each log10 burst intensity bin (*x*-axis). A histogram gate (O1) highlights the singlet population. **d** XY scatter-plot of the blue and green channels on a log10 versus log10 burst intensity plot. Individual populations are gated for statistics. BG1 outlines the single events and BG2 outlines double events. Source data filenames are provided in the Source Data File.

with silicone. These additional modifications, on top of the base design, enabled the device to be space-worthy.

**Results on each sample type**

The on-orbit raw data for the 3 µm diameter pan-fluorescent Flow-Set Pro Fluorospheres (Beckman Coulter, CA) samples are shown in Fig. 5a. The data shows expected results for the five channels: blue, green, orange, forward scatter (FSC), and side scatter (SSC). This particular run lasted 155 s before the end bubble showed up in multiple packets. In the first zoom-in, multiple fluorescences and light scattering bursts are seen on a sample data-trace. Further zoom-in shows a single event, marked by a particle transiting through the two spatially separated 405 and 532 nm lasers. The 405 nm laser is paired with the blue and green channels while the 532 nm laser is paired with orange, FSC, and SSC (Fig. 5b). The spatial separation allows any fluorescence not paired with the specific laser to be excluded, minimizing the need for fluorescence compensation due to spectral crosstalk. The peaks are identified by a threshold and then integrated to the baseline, yielding each peak's burst intensity. Each detector channel has its own set of analysis parameters. The FSC channel has a higher baseline than the other channels due to the nature of FSC, which is on-axis with incident

532 nm laser light, with angles between 0 and 2 degrees masked with a beam block. In contrast to most conventional cytometers, the data collected is all digital, allowing for visualization of all the collected signals and greater flexibility in post-processing without a priori need to optimize instrument settings. Figure 5c shows the analyzed data presented as a histogram of counts versus log10 burst intensity for the Flow-Set Pro Fluorospheres. A single peak is shown, as expected with a percent robust coefficient of variance (%RCV) of 5.84%. Figure 5d shows the XY scatterplot of blue versus green channels. A primary population shows the majority of the beads (BG1 gate) and a second smaller population shows doublets (BG2).

A comparison of data collected preflight on the ground and in-flight on ISS for each detector channel during benchmark commercial cytometer and rHEALTH ONE sample runs with Flow-Set Pro microspheres is shown in Fig. 6a–c. The rHEALTH ONE shows similar single peak populations for each of the channels for both ground and flight runs. The fluorescent channel %RCVs for ground and flight were within 2% of each other, with a lower %RCVs for flight SSC and blue channels (Supplementary Table 5, all runs Supplementary Table 6). The mean bead intensities were similar for green, blue, and FSC channels (±15 detected photons) whereas the orange and SSC channels were brighter

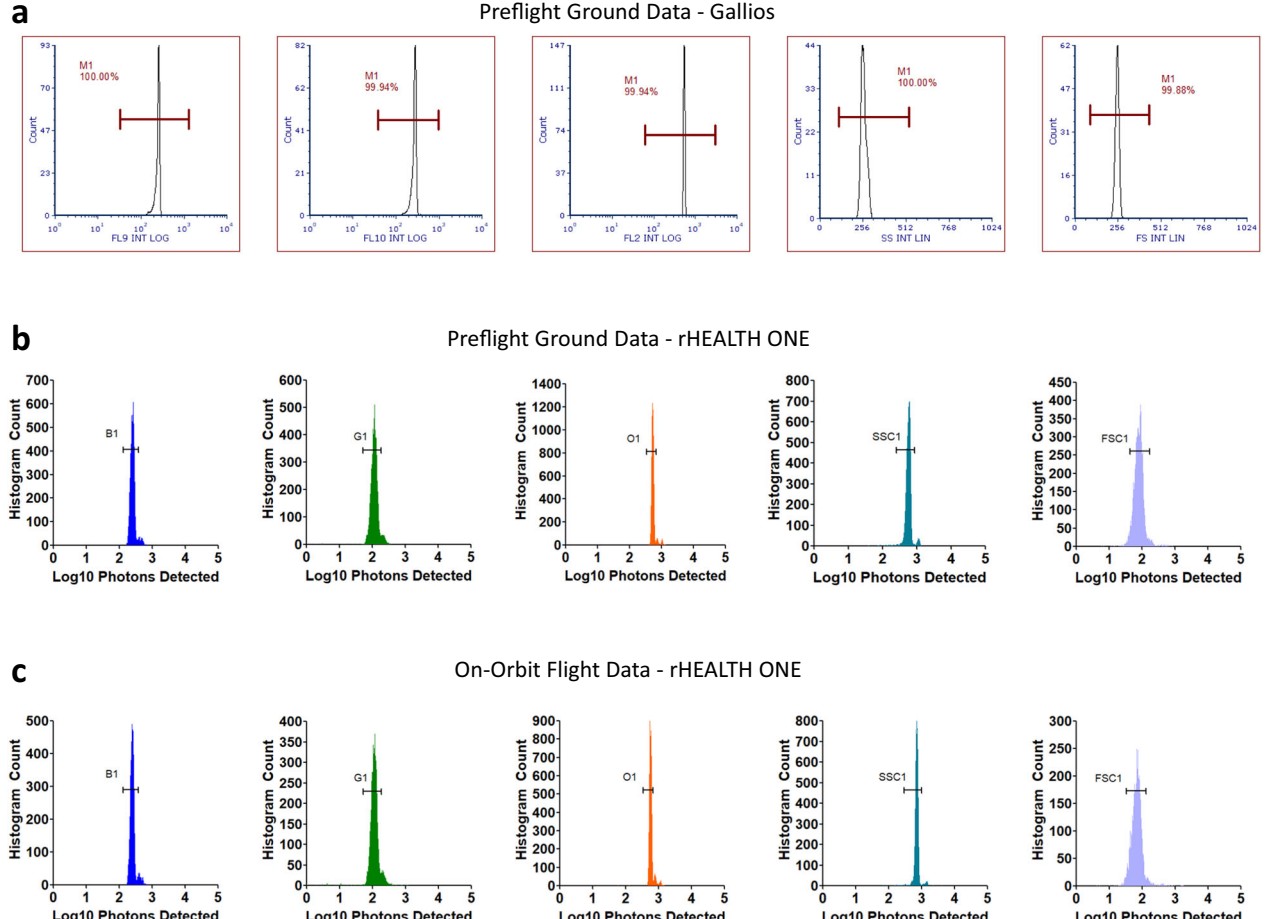

**Fig. 6 | Flow-Set calibration beads data. a** Ground benchmark cytometer data for all five corresponding channels showing, from left to right, blue, green, orange, SSC, and FSC channels. Horizontal gates are shown. **b** Ground rHEALTH ONE data for the same series. Gates are labeled based on their channel and exclude the coincident events. **c** Flight data for the same channel series. The log histograms are base 10. Source data filenames are provided in the Source Data File.

during flight (>+100 detected photons). The flight FSC %RCV was 1.44% greater than ground and also had a slightly higher background noise. The raw counts for each channel are within <1% for each channel to one another for both ground and flight, indicating that each pan-fluorescent microsphere is equally detected in all five channels. Prior to flight, the samples were analyzed with a commercial Gallios cytometer, which has multiple lasers (405, 488, 561, and 638 nm) and comprises a 104 kg, 95 × 61 × 70 cm main unit, a 4 kg 561 nm laser system, and a 30 kg, 72 × 30 × 50 cm supply cart. Similar to the rHEALTH ONE data, the Gallios data show broader histograms for the green and blue channels and a more uniform orange channel. The rHEALTH ONE showed a predominant single peak for each channel as well as the presence of doublets and triplets.

The fluorescence resolution and linearity of the system were tested using differentially dyed pan-fluorescent microspheres with three different intensities. This allowed us to characterize fluorescence measurements of the blue, green, and orange channels. The ground and flight data are shown in Fig. 7a, b. The low, medium, and high fluorescence beads are distinct for each color channel. For each set, a green channel versus FSC scatterplot is included. The scatterplot shows the three populations as well as any coincident events, which can arise when any two beads are in the laser spot together. Supplementary Fig. 6 describes the counting methodology of each of the populations and includes a Bland–Altman difference plot for the counts for each of the populations. The difference plot shows minimal

changes in the relative count numbers, with all counts within −4 to +4% difference, indicating the relative population counts remained consistent. The relative count percentages match well with the Gallios data (Supplementary Table 7, Supplementary Fig. 7, all rHEALTH ONE runs Supplementary Table 8), where the low population is the most abundant population at >33% of all the beads, as measured on both platforms on-ground and on-orbit. The log10 peak burst intensities plotted against the log10 MEF (Molecules of Equivalent Fluorochrome) show linear relationships on all the channels for ground and flight. The dim populations had between 25 and 164 MEFL. The ability to fully resolve the dim population indicates the system maintained high fluorescence detection sensitivity performance on-orbit.

To test the FSC's ability to discriminate between different sizes of microspheres, a mixture of 4, 6, 10, and 15 µm diameter beads (Spherotech PPS-6K) was analyzed. The results of these are shown in Fig. 8a, b (Supplementary Table 9, all runs Supplementary Table 10), which shows the scatter channels. In our system, FSC signal strength increases with microsphere size and measures laser scattering around the microspheres. Forward scattered light between 2 and 20 degrees is collected, whereas direct laser illumination (0–2 degrees) is blocked from reaching the detector. The SSC detector collects light orthogonal to the laser beam (88–92 degrees) and, at these angles, is a measure of particle granularity with less size dependence. XY scatterplots FSC versus SSC show four populations that, when gated, show mean FSC intensity to increase with bead size. The on-orbit data has slightly less

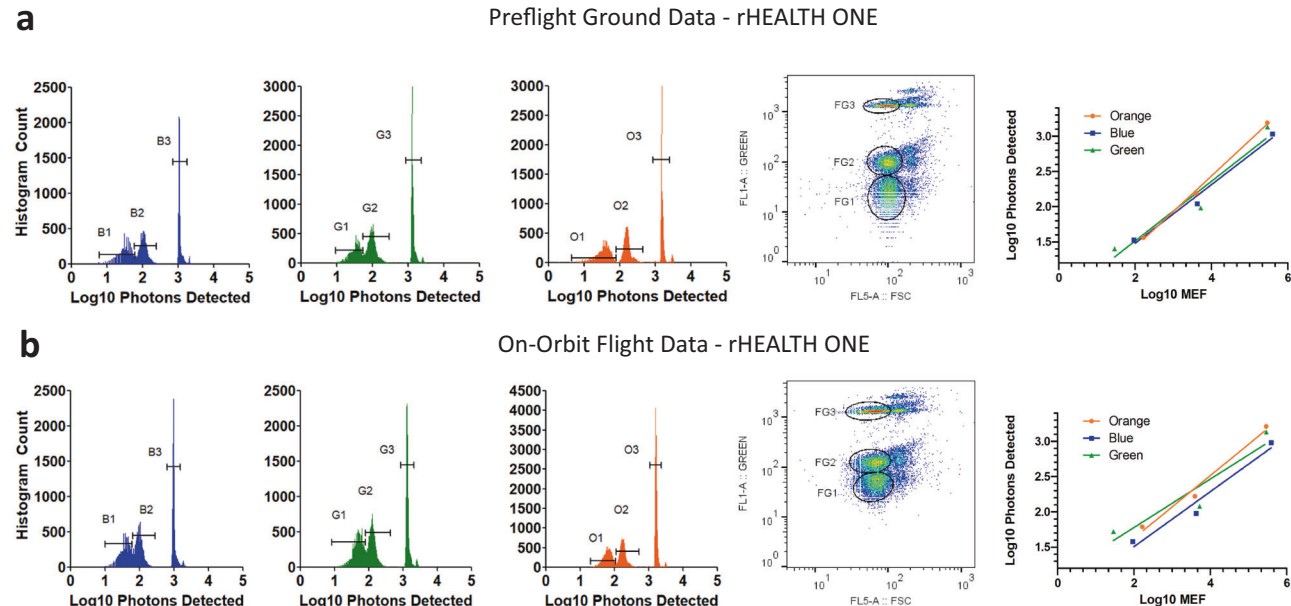

**Fig. 7 | Results for rainbow calibration particles with three different fluorescence intensities, from dim to bright. a** Left-to-right: preflight ground rHEALTH ONE data for log10 blue, green, and orange burst intensity histograms (low, medium, and high populations are identified with gates labeled with 1, 2, and 3, respectively); XY scatterplot of FSC versus green with the singlets gated for low, medium, and high fluorescence with gates ending in 1–3, respectively); and a log10 expected MEF versus log10 photons detected graph. **b** On-orbit rHEALTH ONE data for the same series. Source data filenames are provided in the Source Data File.

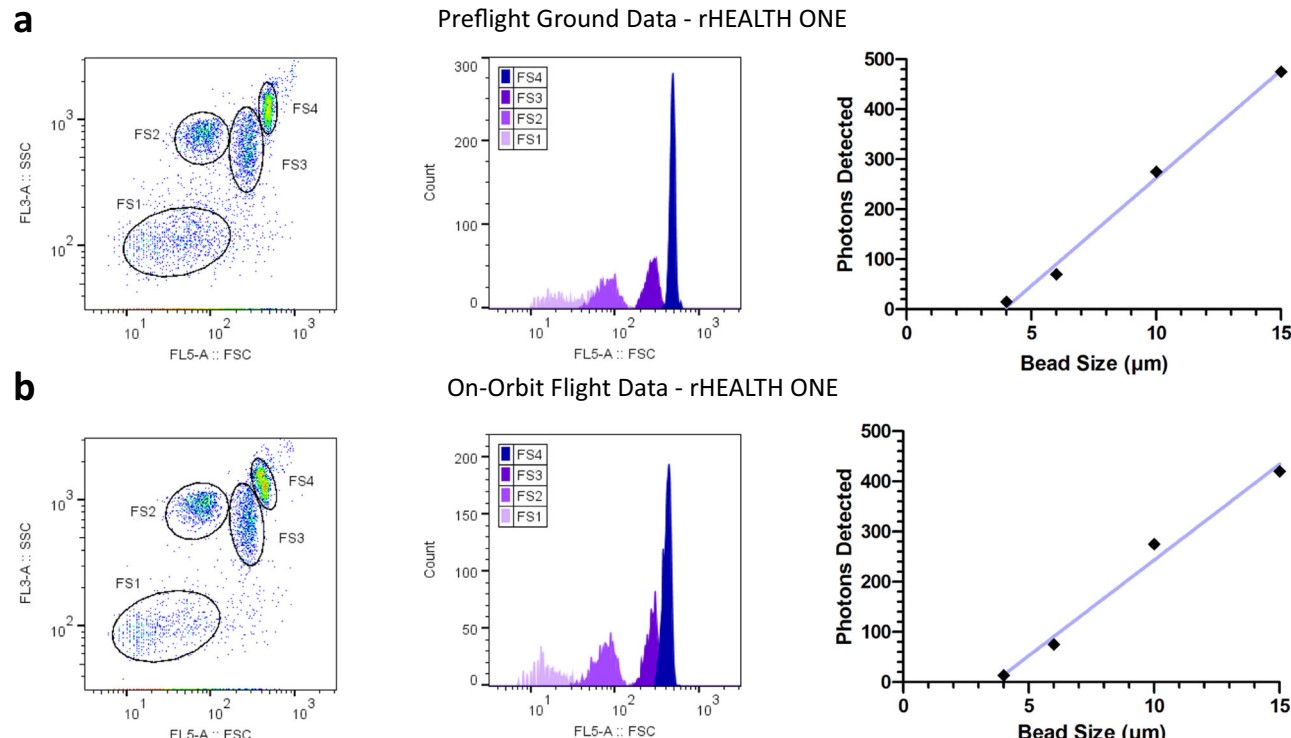

**Fig. 8 | Results for particle size standards of 4, 6, 10, and 15 µm microspheres. a** Left-to-right: preflight ground rHEALTH ONE data for XY scatterplot of FSC versus SSC with gates numbered 1–4 from smallest to largest bead; FSC histograms of the XY scatterplot gated populations; and graph of bead size (µm) versus photons detected. **b** Corresponding rHEALTH ONE on-orbit flight data for a set of four microspheres. Source data filenames are provided in the Source Data File.

separation between 10 and 15 µm microspheres, but more separation between 4 and 6 µm ones. The Gallios data also shows an increasing relationship for FSC in the XY scatterplot, and good separation for the individual beads (Supplementary Fig. 8).

Fluorescent compensation beads (OneComp eBeads, Thermo-Fisher Scientific, MA) were conjugated to fluorophore-labeled antibodies (anti-CD3 V500, anti-CD14 V450, and anti-CD19 PE) to determine the device's ability to resolve multiple colors simultaneously. Both the

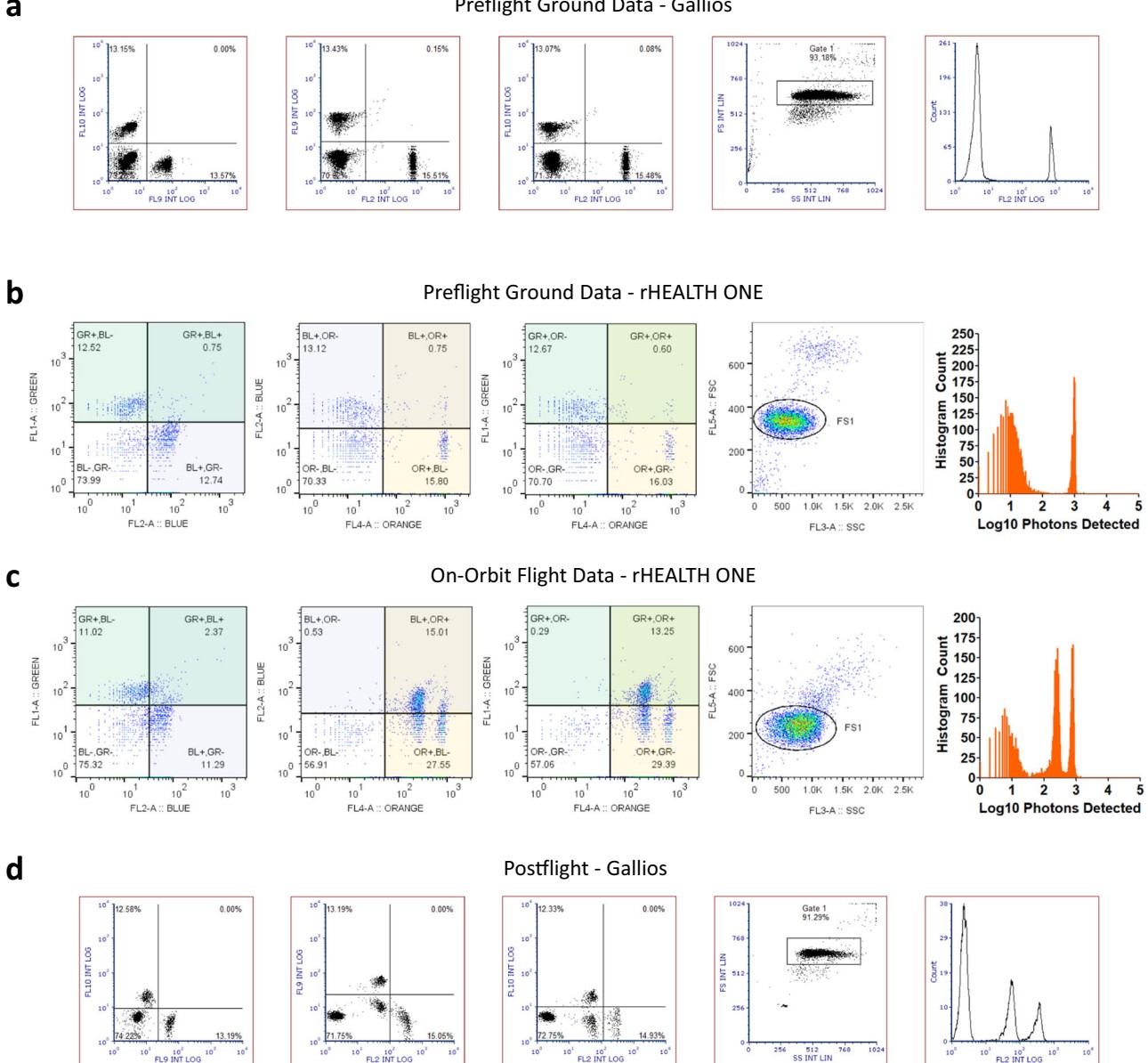

**Fig. 9 | Fluorescence compensation standards with dye-conjugated antibodies, anti-CD3 V500, anti-CD14 V450, and anti-CD19 PE. a** Left-to-right: ground preflight benchmark cytometer data showing scatterplots for blue–green, orange–green, orange–blue, and a log10 histogram of orange FL2 burst intensities. **b** Corresponding rHEALTH ONE preflight ground sample analysis. A transparent color overlay is included to highlight the quadrants that are unlabeled (i.e., BL−, GR −), labeled with a single color (i.e., GR+, BL−), or labeled with both colors (i.e., GR+, BL+). The color channel abbreviations are green (GR), blue (BL), and orange (OR). **c** Corresponding rHEALTH ONE on-orbit flight data. **d** Ground postflight benchmark flow cytometer data. The quadrant analysis is moved to match the percentage of beads seen in the preflight analysis. Source data filenames are provided in the Source Data File.

V500 and V450 dyes are excited of the violet 405 nm laser and the PE with the green 532 nm laser. The ground data shows the correct quadrant-based separation of the three combinations of colors (Fig. 9a, b). The orange channel shows the unlabeled bead population and the single PE-labeled population. During the flight, however, a lower intensity second orange peak unexpectedly appeared, along with increases in orange–green and orange–blue coincidence populations associated with this peak (Fig. 9c). The total number of beads remained similar to ground, including labeled and unlabeled fractions (Supplementary Table 11, all runs Supplementary Table 12). This on-orbit observation prompted the return of the samples back to Earth from the ISS for subsequent terrestrial evaluation. The samples were received 3 months later at Johnson Space Center and post-flight ground Gallios

analysis confirmed changes in the sample measured by the rHEALTH ONE on-orbit (Fig. 9d, Supplementary Table 11). The quadrant analysis shows an overall decrease in the percentage of unlabeled beads, an increase in the orange percentage, and similar proportions of blue and green beads (Fig. 9b, c). When the quadrant analysis is moved to the right to include the new dim orange peak, the starting percentages are recovered (Fig. 9d).

A total of 32 runs (4 blank and 28 samples) were performed across the 2 days of on-orbit operation (Supplementary Table 13). Excluding the blank runs, we were able to perform double the number of planned runs (28 versus 14) since sample running required less time than anticipated. Of these runs, 26/28 yielded good-quality data in all channels. One sample without data were a designated practice run on

**Table 1 | Instrument resource utilization comparison**

| Instrument resource utilization comparison | | | |
|---|---|---|---|
| Metric | rHEALTH ONE | Gallios | Fold reduction |
| Volume | 2,808 cm³ | 513,650 cm³ | 183× |
| Power | 2.9 W | 1,500 W | 517× |
| Mass | 1.5 kg | 138 kg | 92× |
| Sheath volume | 0.060 L | 10 L | 166× |

The resource utilization of the rHEALTH ONE and benchmark Gallios cytometer is listed as well as the fold reduction in resource utilization.

the first day of operation. An incomplete run also occurred when the sheath fluid ran low. The runs had an average duration of 143.10 s and an average of 71.55 million raw data points.

## Discussion

A microgravity-capable microvolume flow cytometer demonstration unit, a spaceflight-modified rHEALTH ONE, was demonstrated on the ISS. The device utilized 517× less power, 183× less volume, 92× less mass, and 166× smaller sheath reservoir than the ground-based benchmark Gallios cytometer (Table 1). This low resource utilization allowed it to be launched into space. It was tested over two days on-orbit with samples that characterized its performance. More sample runs were performed than originally planned (Supplementary Table 14). The device was able to meet the predefined criteria for a successful hardware technology demonstration (Supplementary Table 15): data collection in all five channels, greater than or equal to three runs per sample, demonstration of sheath-based hydrodynamic focusing, complete analysis of each sample, and data comparable to ground data.

The successful operation of the device highlighted several important technical developments that enable single-drop cytometry in microgravity. The use of all solid-state lasers and detectors with fixed alignment optics allowed the achievement of a highly miniaturized cytometry module that maintained sensitivity to dim events. This module tolerated the complex vibration and g-profiles on rocket launch and subsequently, once onboard the ISS, allowed the collection of cytometry data simultaneously from the five photon counting detectors in microgravity. The microgravity loading procedures were used to minimize the amount of air bubbles in the system, and the engineering of the fluidic system allowed sheath flow-based hydrodynamic focusing, the standard of conventional cytometers, to be performed. An in-line microvolume sample loader that worked with a microgravity wicking procedure enabled the repeated loading of microvolumes of sample into the device, without any excess unused volume.

The rHEALTH ONE ISS demonstration advanced the current knowledge about performing flow cytometry in microgravity. Small drops of sample, that were manifested separately in dropper bottles, were wicked into sample consumables and successfully analyzed over the course of multiple on-orbit runs with the rHEALTH ONE device. This is in contrast to the Microflow1, where cartridges had to be pre-loaded with a larger volume of sample (1.6 mL) prior to flight and were susceptible to micro-bubbles which rendered some of the samples unusable[29]. The rHEALTH ONE device increases the possibility for the inclusion of cytometry capabilities during space exploration missions by offering a miniaturized, free-space laser optics approach that offers greater flexibility than the Microflow1's integrated fiber-optic flow cell. The Microflow1's fixed geometry limited the total number of fluorescent channels and prevented the addition of the critical FSC channel. The rHEALTH ONE's cytometry module allowed for a second laser, and two more detector channels, including FSC. The rHEALTH ONE demonstrates sheath-flow hydrodynamic focusing cytometry in microgravity. To do this, a fluidic system was developed to minimize

bubble interference and to control microvolume sample loading. The benefit of a sheath-based system is the one-by-one delivery of cells and particles through the laser excitation region. The sheath-based system allows the sample to be pulled away from the wall of the analysis channel, which increases sample velocity uniformity and decreases the risk of cell aggregation at the zero-velocity boundary condition of the channel wall. The low pressures used to push the fluids through the system (70 mbar) minimize the required amount of filtered water sheath per run. The minimal, best-case amounts of required sheath water include 2.44 mL for the startup prime and 1.13 mL for each subsequent run (Supplementary Table 16). The use of water as the sheath allows the approach to be compatible with potable water sources on spacecraft, as long as it is adequately filtered at the level used in the experiments.

Future improvements or additions are envisioned that can further improve or augment performance. Over an hour of each of the test sessions was used for filling the fluid bags and removing the air bubbles, the success of which is operator-dependent and where improper filling can result in undesired air bubbles. Pre-filled, gas-impermeable fluid bags could streamline instrument operation. Additionally, the instrument setup requires manual priming of the system through the burp port. While this maneuver is short in duration, it could be automated to improve usability, especially since it is required during device startup after having been stowed, facilitating start-up after launch or between extended on-orbit uses. Incorporation of even more rigid optical elements will be important since a slight movement likely resulted in the higher FSC channel noise observed on-orbit compared to on the ground. Biological test samples could be evaluated in the future. This would require meeting NASA's biohazard containment constraints and also necessitate the use of microgravity-compatible sample preparation devices, such as the easy-to-use Whole Blood Staining Device[44], or automated microfluidic methods[37], both of which utilize sample volumes compatible with capillary blood sampling. Assay capabilities can be expanded to include key tests envisioned for exploration missions, including blood chemistry, blood counts, cardiac biomarkers, urine analysis, liver function, kidney health, and coagulation[45]. The software could be improved with automatic processing of raw data into burst intensities and the addition of features familiar to flow cytometry scientists such as user-friendly approaches for gating, thresholding, compensation, detector voltage adjustment, and laser intensity control. The data analysis could be upgraded to provide results in a readily interpretable format for astronaut users, especially for cell counts, cell subpopulations, cell parameters, highly-multiplexed biomarkers with differentially-dyed microparticles[23] or nanostrips[26], and other test panels used to guide clinical decision-making.

The ability to analyze biomedical samples via in-flight lab analysis throughout a mission has been a long-standing aim of NASA's Human Research Program[46]. The vast majority of our understanding of space medicine biomarkers, blood cell changes, immune function, and cell-based biology is derived from downmassed samples, which require a long journey back to Earth prior to analysis in a central lab. Samples may degrade, change, and become unreliable during transit. This was observed with the fluorescent compensation beads, which was the only sample used in the rHEALTH ONE ISS demonstration that had a biological component to it. Varying sample storage conditions likely led to the observed changes in the sample. This could have been from room temperature sample storage en route to and onboard the space station and/or from the higher doses of radiation (1 mSv per day on the ISS[47] versus 2.4 mSv per year on Earth[48]). Either of these could have led to the desorption of the antibodies and subsequent reattachment to the unlabeled fraction of the fluorescent compensation beads. Ionizing radiation could have altered the surface charges on the beads and non-refrigerated temperatures could have increased the desorption kinetics. Protein degradation could have also been accelerated.

High $g$-loads and vibration during transit is less likely a possibility given that laboratory samples typically are subject to mixing and vortexing without issues. Our challenges with this sample have been observed with other space-based biological samples. For instance, the harsh transit conditions during sample transport for the NASA twins study resulted in the loss of telomerase activity from samples[14].

On-orbit biomedical analysis would aid a more complete understanding of spaceflight biology by providing timely information on freshly acquired samples. Flow cytometry was used as the core analytical modality since it has versatile and diverse applications, ranging from blood cell counts, immunophenotyping, multiplexed biomarker assays, bacteria/virus analysis, and general particle sizing (which could be used for lunar or Martian dust). Given the highly limited in-cabin resources and the inability to rely on Earth for analysis support or resupply, a single instrument that can achieve the greatest assay diversity and multiplexing is desirable. Aside from research studies, a microvolume cytometer that can analyze self-collected capillary samples can guide critical preventative and emergent medical decision-making. Sample return challenges become exponentially more daunting, if not impossible, as we perform missions that return us to the Moon and travel deeper into space to Mars. These challenges of space are also analogous to those on Earth in minimally resourced settings such as developing countries, satellite labs, pharmacies, and homes, where point-of-care analysis is desirable. The rHEALTH ONE ISS demonstration provided a step forward in realizing immediate and actionable biomedical information in environments where no cytometer has gone before.

## Methods

### Flight sample preparation
All samples were prepared in sterile conditions, bottled separately as 1 mL of solution in 3 mL dropper bottles (United States Plastic Corp. P/N 66529), protected from direct light, and stored at 2–8 °C for longest shelf-life before being delivered for launch at ambient temperature. Samples A: OneComp eBeads, B: Spherotech PPS-6K, and C: Spherotech 3-peak fluorescent standards were prepared by the JSC Immunology lab. Sample D: Flow-Set beads were prepared at ZIN technologies since it was a simple process and reduced shipment of materials. Samples were diluted to a concentration of beads (polystyrene microspheres) that allowed the rHEALTH ONE analyzer to be set to one target pressure (1 psig, 70 mbar) throughout testing.

For sample A, 5.0 mL of OneComp eBeads (Thermo Fisher P/N 01-1111-42, Lot 2297369) ~4 μm in diameter was stained in three separate batches with one color of fluorophore-conjugated antibodies each, washed thoroughly, then resuspended together in 20.0 mL cell culture grade water (Sigma–Aldrich W3500-100ML, Lot RNBK3069). The staining ratio was 50 μL OneComp eBeads (5,000,000 beads/mL) to 5 μL antibody for a final dilution of 1:11 for each antibody. Fifty microlitres of Food Color & Egg Dye–Blue (McCormick UPC 52100071077, Lot FEB 10 25 H 03:48) was added, and then 1 mL of the final solution was transferred into each dropper bottle. Antibodies used for staining (all are mouse IgG1, kappa isotype control): CD19 PE (Tonbo Biosciences P/N 50-0199-T100, Lot C0199110320503, 50 μg/mL), CD14 V450 (Tonbo Biosciences P/N 75-0149 T100, 100 μg/mL, Lot C0149092019753), CD3 V500 (Tonbo Biosciences P/N 85-0038 T100, Lot C0038012221853, 100 μg/mL). Each drop of beads has a positive population that captures the mouse antibodies and a negative population that does not react with the antibodies.

For sample B, 1.0 mL of each size–4, 6, 10, and 15 μm in diameter–from the polystyrene bead particles, size mix (Spherotech P/N PPS-6K, Lot AM02) were mixed then diluted by the addition of 16.0 mL cell culture grade water (Sigma–Aldrich W3500-100ML, Lot RNBK3069). Fifty microlitres of Food Color & Egg Dye–Blue (McCormick UPC 52100071077, Lot FEB 10 25 H 03:48) was added, and then 1 mL of the final solution was transferred into each dropper bottle.

For sample C, 5.0 mL of Rainbow QC calibration particles, three peaks (Spherotech P/N RQC-30-5, Lot AL01) were diluted by the addition of 15.0 mL cell culture grade water (Sigma–Aldrich W3500-100ML, Lot RNBK3069). Fifty microlitres of Food Color & Egg Dye–Blue (McCormick UPC 52100071077, Lot FEB 10 25 H 03:48) was added, and then 1 mL of the final solution was transferred into each dropper bottle.

For sample D, 2 μL of Food Color & Egg Dye–Blue (McCormick UPC 52100071077, Lot FEB 10 25 H 03:48) was added to each dropper bottle then 1 mL of Flow-Set Pro Fluorospheres (Beckman Coulter P/N A63492, Lot 3941176 F) at full concentration was transferred into each dropper bottle.

### Sample loading and running
After a gentle inversion process, the individual test samples in dropper bottles were dispensed onto polyimide tape adhered to the work surface. The rHEALTH ONE sample consumable tip was tapped against the drop to fill it by capillary action. The presence of blue dye in the samples allowed visualization of a proper 10 μL capillary fill. The door to the rHEALTH ONE was actuated to allow the sample consumable to be loaded. Closing the door forms seals around the end of the sample consumable via gaskets, allowing it to be in line with the fluidic system. The sample loading mechanism forms a fluid-fluid interface at the leading edge of the consumable and an air-fluid interface at the trailing edge. Pressure-driven flow at 70 mbar (1.0 psig), actuated through the *rHEALTH Capture v59e4* was utilized to drive the sample through the device's optical block. The full 10 μL of sample was analyzed within 3 min and the files were stored in a TDMS format suitable for *rHEALTH Viewer v.26.2f_exporthistogram* analysis. FCS files were exported for analysis and visualization on FlowJo (Becton Dickinson, OR).

### rHEALTH ONE device description
The base rHEALTH ONE device (purchased from rHEALTH Inc., MA), developed with NASA support by authors from DMI and rHEALTH, has its specifications listed in Supplementary Table S1. The device measures $13.4 \times 17.8 \times 13.0$ cm and is 1.5 kg. Power and data are supplied with a USB 2.0 port on the back of the device. Pressure to the vials is provided by a small DC-powered eccentric diaphragm pump inside the unit. The lasers are 405 nm violet 5 mW and 532 nm green 20 mW lasers. Each of these has a rate of >5000 h of operational life. The sample consumable allows a minimum of 5 μL per sample and up to 10 μL (as demonstrated on the ISS). The sample flow rate can be adjusted by changing the pressure on the device and it can range from 2 to 10 μL per min. As operated on the ISS, the sample flow rate is approximately 3–4 μL per min, which is considered a very low flow rate. The system on the ISS operated with 70 mbar to achieve this low flow rate. Low-pressure operation allowed minimization of the use of fluids. The particle throughput can be up to 1000 events a second. As operated on the ISS, the event rate was below 100 per second. The event rate is dictated in part by the sample concentration, which can range from $10^4$ to $10^7$ particles per mL. The unit has a dedicated in-line, zero-dead volume sample loader. The fluidic system is rinsed with the fluid in the cleaning bottle after each use. On the ISS, the cleaning compartment was water to remove the need for double containment of the device. The device supports up to a five-log assay dynamic range and has two software modules, the *rHEALTH Viewer* for visualization and the *rHEALTH Capture* for data capture and device operation. The data output is all digital, which is different than analog-based cytometers. This allows for rethresholding and also changing analysis parameters. This flexibility bypasses the need to optimize run parameters prior to runs and offers additional thresholding capabilities after the runs. The device supports the use of nanostrips, lumibeads, multiplexed microspheres, and cell-based assays. On the ISS, contrived samples had to be utilized to avoid the need to address biosafety considerations.

The optical module is mounted on the inside wall of the unit, in close proximity to the sample loader to minimize the transit time of the sample to the flow cell. This mounting configuration also allows the optical module to be cooled by a fan placed at the base of the instrument. This provides cooling to the heat sinks, which are thermally coupled to the optical module and both lasers. The fan intake is from the bottom, blowing up through the top of the hinged bottle assembly. The lack of buoyancy in zero gravity necessitates fan-based cooling since warm air does not rise to the top. No peltier-based cooling is needed if the device is operated within the bounds of 15°–35 °C. The system requires warm-up to stabilize the lasers. The heat generated by the PCBs, lasers within the enclosure allows a steady-state to be attained after 20 min of power applied to the lasers. On the other side of the unit is a fluidics module that has integrated low-power solenoid valves and a two-stage pressure regulator that is motor-controlled. Closed loop feedback with a pressure sensor allows for precise pressure regulation.

The electronics inside the unit consist of multiple custom PCBs, including a main, detector, and LED PCB. The main PCB is at the bottom of the unit and manages the control of the device, including valves, fan control, pressure sensors, and data. A Microchip SmartFusion System on a Chip (SoC) with FPGA controls the primary functions, including data transfer from the detector PCB to the main board and from the main board to the PC connected via USB. Wires connect the main PCB to a door sensor, which determines the door state (open or closed). The optical module has its own separate detector PCB, which is mounted in close proximity to the silicon photomultiplier detectors (SiPMs, Hamamatsu, and JP). The detector PCB has its own microcontroller, a dsPIC33 operating at 70 MHz with 128 kb program memory size, four direct memory access (DMA) channels, multiple analog-to-digital converters (ADCs), and pulse-width modulation (PWM) channels. This dsPIC33 takes data coming from the SiPM and shuttles them to the main board. This allows the operation of the detector board at the desired data rates of 10 µS bin intervals. The five SiPMs are supplied from a common high voltage. The individual channels have their bias voltage set by adjusting the positive input to the opamp, so that the actual bias voltage is the common high voltage minus the adjustment voltage. The output is a negative outgoing pulse that is AC coupled to the gain stage. The SiPMs are capable of detecting single photons and they can self-quench within about 60 ns. The SiPMs are mounted on individual flex circuits to allow each detector to be individually aligned. Each flex board has a separate pre-amplifier. The pulse output from the preamps is further amplified with an opamp set for a gain of ten. These amplified signals are then fed into comparators, and the outputs of the comparators are used as triggers for counters within the dsPIC33. The counts are stored with sample periods of 10 µS, which is reset every 10 µS for continuous measurement. The counts are stored as a single byte of data for each channel. This data are sent to the main PCB in real-time over an SPI channel, allowing collection and real-time visualization of this data on the PC using the *rHEALTH Viewer*. The detector PCB also contains laser driver circuits that allow the output of the lasers to be adjusted and monitored. The laser outputs are controlled by an active circuit that adjusts the current passing through each laser diode. Each laser also has a thermistor mounted on its flex board to track the temperature of the laser and correct brightness variations with temperature.

The optical module has two compact, commercially available diode lasers (405 nm 5 mW, Egismos, Taiwan, and 532 nm 20 mW Snake Creek Lasers, PA). These are at right angles to each other and combined using a dichroic filter. The collimated lasers are shaped into elliptical laser spots (200 × 10 µm) by a pair of cross-cylindrical lenses. An achromat focuses the lasers into the rectangular fused silica flow cell (Hamamatsu, JP). The laser spots are offset by 400 µm (approximately two channels' width) to decrease spectral crosstalk and

decrease the need for fluorescence compensation. Fluorescence and SSC is collected by an integrated lens that is fabricated with the flow cell. This allows for high numerical aperture light collection. This light is directed to a set of dichroic and bandpass filters that color separate the three color channels and the SSC laser light. This light is focused on the SiPMs connected to the preamp flex boards. The FSC channel has separate optics and light collection. A neutral density filter and beam block attenuate the direct laser light, allowing it to be measured also by a photon-counting SiPM.

Several design considerations made this optical module tolerate the vibration and *g*-profiles during launch and also in zero gravity. These include the low mass of each of the components in the module. This minimizes the amount of force exerted on fasteners and epoxy joints. None of the components were adjustable. This in contrast to conventional cytometers that generally have screw adjusters and motion control for aligning the system in the field. The use of all solid-state components meant the greatest level of miniaturization possible. The conventional photomultiplier tubes have a photocathode with multiple physical stages of anodes and dynodes for signal amplification. In SiPMs, this amplification is within the silicon. The flow cell is integrated into the optical cytometry module. This was possible since this flow cell and connectors were miniaturized with a custom flow cell top fitted with hypodermic gauge pins. This approach minimizes fluid volume usage and air bubble trapping within connectors and allows for integration with the optical module. Typically, the flow cell is separate from the optics in larger cytometry systems, but in our system, because of the fixed alignment approach, it necessitated an integration with the module.

### Specific device modifications required for spaceflight

A number of modifications were made to the rHEALTH ONE device to ensure safe and functional operation in the microgravity environment aboard the ISS. To address any potential fluid leaks from the fluidics system, 1.80 mm wide zip ties (PLT.6SM, Panduit, IL) were used to secure polymer tubing to pins and barbed connections. Additionally, any plastic tubing-to-tubing connections were strengthened with 3.175 mm wide, adhesive-lined 3:1 heat shrink tubing. To safeguard the PCBs against water damage, DOWSIL™ 3140 clear RTV was thinned with an epoxy thinner (xylene) and applied to the PCBs using a high-volume low pressure, gravity-fed spray gun. DOWSIL™ 3145 gray RTV silicone was dispensed onto all fasteners, threaded parts, and PCB connections for additional strength to withstand the forces/vibrations at launch and return. This same epoxy was also used to seal the optical block and reduce any dangerous laser light leakage. Two Velcro strips were applied to the device's lid to prevent unwanted movement in zero-*g*. To keep the device stationary during mission operations, standoffs with Velcro on the underside were adhered to the bottom of the device.

### Fluidics for zero gravity bottles, waste, and reservoir

Several modifications were made to the external fluidics and procedures were developed to separate, direct, and contain the air and liquid during transfer and operation. Tubing lines that handled liquids were adapted to Luer lock connections that would be securely connected to mating Luer lock bags and plastic syringes but also easily connected and disconnected by the crew. A Luer lock bag with a self-sealing valve (Origen PL120-2G that includes an Origen NFV) was used to transport the liquid (Sigma–Aldrich W3500 cell culture grade water) needed for operations to the ISS and the same part was used to replace the waste bottle in capturing liquid waste. ISS safety requirements require the liquid to remain contained, not free-floating in the cabin. A self-sealing Luer lock (Origen NFV) was added to the priming line and tubing extensions were added to both priming (IDEX P-850, Masterflex 06407-71, IDEX P-857, and Origen NFV) and waste (IDEX P-857,

Masterflex 06464-90, and Masterflex 30505-92) lines for maneuverability during operations and visibility of the flow of liquids and air bubbles. The fluid bag assembly inside the bottles was made from two medical balloons (Nordon Medical 20005500CA) with walls at 0.014 mm thick, but durable across thousands of inflation/deflation cycles. These were connected by a barbed Y connected (IDEX P-860), 1/8" OD 1/16" ID Tygon tubing (Masterflex 06407-71), adapter (IDEX-P-857), and Origen NFV. This design directly utilized as much of the bottle volume as possible (requiring less refilling) while still being compressible by 70 mbar (1 psig) of air, small enough to pass through the bottleneck and accommodate the necessary Luer adapters, and flexible enough to be filled and connected through a complex sequence of steps over multiple uses without leaking. The original filters were removed and modified to allow in-line Luer lock connection (IDEX P-235, IDEX P-200, and IDEX P-675) between the device and each bag assembly.

### Device and sample transport

The unit was purged with air and packed dry for transport. Samples were prepared on 2021-10-29 and 2021-12-02, kept refrigerated at 4 °C then transported at ambient temperature starting -2021-12-20 for launch, operations, and return to ground. Launch was 2022-02-19 and operations 2022-05-13 and 2022-05-16. Samples were returned at ambient temperatures on 2022-08-20 and analyzed at NASA's Johnson Space Center. The rHEALTH ONE analyzer and the water bags were all packed dry for launch to avoid developing fluid bubbles. Filling the rHEALTH water bags on the ground and pre-loading them in the rHEALTH bottles was considered but due to the permeable materials, they would develop air pockets that would require removing and refilling them on-orbit, negating the benefit.

### Payload development

Payload development followed NASA's experiment flight hardware development process, for modified COTS devices. In order to qualify for this approach, the device had to be commercially available. This included hazard and safety analysis unique to the payload to determine operational requirements and containment then verification to ensure the requirements were met. This included vibration testing according to launch loads to verify the lasers and frangible material inside the device remained contained and required reducing the EMI over the specified ranges reserved for ISS systems by applying copper tape. Specifically 0.04 mm thick, RF EMI shielding copper foil PSA tape (Parker Chomerics, CCH-18-101-0100) was applied to the interior of the unit's housing. The fan opening was covered with a 150 per 2.54 cm, stainless steel mesh screen to complete this Faraday cage. Sections of the PCBs that came in contact with the housing were additionally protected with Kapton® tape to insulate against electrical shorts. A 1-m-long ground wire was secured to the inside of the device for adequate grounding.

Further modifications and controls set in the procedures ensured all fluids would remain contained according to the level of containment required by their NASA toxicology and biohazard assessment. Since activities required great dexterity and visual acuity with many small parts and clear materials, it was decided to tailor the experiment to allow crew operations to occur in the cabin instead of a glovebox. Water as the sheath and cleaning fluid and TOX 0 samples were used to reduce the biohazard risk to the crew. Cell culture grade water (de-ionized and sterile) and sterile samples were chosen to prevent the occurrence of clogs such as from minerals or biofilms. Biological samples such as blood or saliva were not used, which would have added uncertainty to the measurements and needed greater than TOX 0 cleaning fluid after running. This demonstration focused on the analyzer's ability to function in microgravity, testing the underlying fluid dynamics of sheath-based focusing in microgravity, and its performance compared to ground.

### Crew training

Crew training was only for the data collection portion of using the device (no sample preparation, no adjustment of settings, no *rHEALTH Viewer* user processing of the data, etc) and was a 30–45 min period to review the summary PowerPoint, the procedures, and the short videos of main steps. Payloads are designed to minimize crew time required, including training. The crew was guided through all steps with the majority of the work done by the ground team (samples tailored to the device, sample preparation, tailored procedures, real-time verification, and troubleshooting, post-run data processing and analysis, etc.).

### Data analysis

The data analysis was performed using the *rHEALTH Viewer v26.2f_exporthistogram* (rHEALTH, MA) with peak calling settings that were preset for each of the samples. The same peak calling parameters were utilized for both the ground and the flight data to avoid any inconsistencies. The peak files were exported in flow cytometry standard (FCS v3.1) for further visualization, gating, and statistics in FlowJo v10.8.1 (Becton Dickinson, CA). Graphs were created either in FlowJo or in GraphPad Prism 9.5.1 (Dotmatics, MA). The collected data were organized into tables for further analysis using Microsoft 365 Excel (Microsoft, WA) and Google Sheets (Google, CA).

### Statistics and reproducibility

The samples were selected to test the flow cytometry performance of the rHEALTH ONE for its ability to resolve fluorescence intensity, microsphere size, fluorophores, and bead populations. At least $N = 3$ triplicates of the four samples were analyzed. The samples were blinded to the authors at DMI and rHEALTH until the day of the on-orbit operations. Two runs were excluded from the analysis: one was a practice run that did not yield any data, and another was an incomplete run that resulted from running out of the sheath. The total number of samples was limited by astronaut crew time and the 2 days of operation was designed to attain at least a minimum of triplicates. No statistical method was used to predetermine the sample size. The sequence of the sample runs was not randomized.

### Reporting summary

Further information on research design is available in the Nature Portfolio Reporting Summary linked to this article.

## Data availability

All data supporting the findings of this study are available within the article and its supplementary files. The rHEALTH ONE FCS source data for the ground and flight tests are available via FlowRepository[49], under accession FR-FCM-Z76L. Source data filenames are provided in the Source Data File. Any additional requests for information can be directed to, and will be fulfilled by, the corresponding authors. Source data are provided with this paper.

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

## Acknowledgements
DMI (DR, SB, ZS, JC, RY, DE, EC) was supported with NASA contract 80NSSC18C0162. KBR (RM) and ZIN Technologies (RV, KC) was supported by NASA Contract NNC14CA02C. The authors would like to thank the Exploration Medical Capability Element of the NASA Human Research Program for their support and contributions to this technology demonstration effort.

## Author contributions
D.R. performed final assembly, device testing, and data analysis. R.M. performed device testing and developed fluid bags, flight protocols, and data analysis. R.M. and R.V. performed flight qualification and protocols. S.C. performed the on-orbit experiments. Z.S. assembled and tested the optical cytometry module. J.C. graphed the data. R.Y. performed device testing. K.C. managed the flight project. S.B. wrote the *rHEALTH Capture* and *rHEALTH Viewer*. D.E. designed the electronics and wrote the device firmware. B.C. developed the sample protocol, formulated the flight samples, and performed benchmark testing. E.N. reviewed the project, B.L. reviewed the project, G.P. reviewed and managed the project, E.C. designed the device and wrote the paper.

## Competing interests
E.C. is an equity holder of DMI, rHEALTH and listed as an inventor on US Patents 9194780, 9617383, 9568425, 9835542, D785781, 10180442, 10279347 B2 related to the technology. The remaining authors declare no competing interests.
