## [Peer Review File · Nature Communications]

REVIEWER COMMENTS

Reviewer #1 (Remarks to the Author):

This paper introduces a compact flow cytometry system designed for real-time lab analysis in microgravity conditions, with the remarkable potential to provide diagnostic information for astronaut health management during spaceflight, negating the need for Earth-based analysis. However, the proposed technology appears to offer only incremental improvements, such as minor miniaturization of existing flow cytometers, the use of positive pressure for sample injection, and copper shielding to protect against space electromagnetic waves. Furthermore, the expected technical impact seems to be limited, especially considering the availability of portable cytometry products with similar or even superior performance already in the market, like CyFlow Cube6 from Sysmex and Moxi Go II from Orflo. For these reasons, it seems reasonable for this paper to be published in a space-related specific journal. My additional comments are as follows:

- (1) In space, electromagnetic wave distribution can vary significantly depending on time and location. For this reason, long-term signal monitoring in units of days, rather than seconds, seems necessary.
- (2) When experiments are repeated, the data presentation should include the standard deviation to provide an intuitive understanding of the degree of deviation. For example, in Figure 7, the authors should add a graph showing the deviation for each test result within each bead population.
- (3) Figure 7 shows that the sample was degraded during the experiment. Is this damage attributed to the storage environment in space? Please provide further discussion on this issue.
- (4) The authors should provide a representative plot of SSC vs FSC for cytometry data in Figure 7.
- (5) Clinical tests seem to be limited by NASA regulations, but testing should be further conducted on clinically valuable samples.

Reviewer #2 (Remarks to the Author):

The authors present a compact flow cytometer with an extremely small footprint and whose performance matches with those of much larger, desktop commercial cytometers. The miniature cytometry-based analyzer, the rHEALTH ONE, is based on sheath-flow, uses sample volumes as small as 10uL and collects data on five separate channels simultaneously using a 10us bin interval. The FSC, SSC and the fluorescent signals compare well with those obtained using a standard cytometer thereby demonstrating the precision and accuracy of the measurements. The device was tested in zero-gravity conditions and performed well. These are excellent results and mark an important step in cytometry design.

Having said that the manuscript lacks design details, which makes reproducing their results impossible. For example, it is not clear what design/scientific innovations enabled miniaturization of the optical unit. What was the design for the sheath-flow system, especially the flow cell where the laser beam strikes

the particles? What optics were used to shape the beam - was the cross-section of the laser beam, flat-hat or elliptical? What lasers were used in the set-up - were these bought off the shelf? What was the power rating and stability of the lasers? How were the lasers cooled since it is well known that laser heating leads to instability of the beam intensity? What type of photodiodes were used - were these bought off the shelf? Commercial flow cytometers use photomultipliers due to the low intensity of fluorescence. Are these photomultipliers or avalanche photodiodes or standard photodiodes? There are no details of the sensor circuit - what opamps were used? What was the amplification? What was the signal to noise ratio? What type of microcontroller was used to capture signals? What design aspects enabled the device to withstand high-g forces? Without these details it is not possible to assess the scientific/design innovation that went into building the compact design. Again, these details are needed to reproduce the results reported in the manuscript, without which the manuscript is not suitable for publication in a scientific journal. As an example of a scientific article on design of cytometer, I suggest reading the paper by Habbersett et al (Cytometry Part A 71A: 809-817, 2007) who give full details of their cytometer including all components and circuits.

My second major criticism is regarding their FSC results. They state that the FSC signal strength was found to be proportional to the particle diameter. This result is not correct. Mie theory predicts the FSC intensity to be proportional to the square of the particle size (or equivalently, particle cross-sectional area) - see "Absorption and Scattering of Light by Small Particles", Bohren & Huffman, Page 404 and Fig 13.10. These predictions have been confirmed via experiments (see for example, Pinnick and Auvermann, "RESPONSE CHARACTERISTICS OF KNOLLENBERG LIGHT-SCATTERING AEROSOL COUNTERS", US Army Electronics Research and Development Command, 1979).

Based on the above observations, I am not inclined to recommend the manuscript for publication in Nature communications.

Reponses to Reviewer Comments for:

“Single Drop Cytometry Onboard the International Space Station”

REVIEWER COMMENTS

Reviewer #1 (Remarks to the Author):

Reviewer #1 makes several good points and these are addressed below in purple. This paper introduces a compact flow cytometry system designed for real-time lab analysis in microgravity conditions, with the remarkable potential to provide diagnostic information for astronaut health management during spaceflight, negating the need for Earth-based analysis. However, the proposed technology appears to offer only incremental improvements, such as minor miniaturization of existing flow cytometers, We included a more comprehensive description of the rHEALTH ONE device per Reviewer #1 and Reviewer #2, the use of positive pressure for sample injection, and copper shielding to protect against space electromagnetic waves. Zero gravity has no buoyancy so any air trapping in the cytometer has to be eliminated. In this environment, air bubbles remain in the middle of the fluid solution. On Earth, air bubbles present issues to fluidics and in space, this is an even much more significant issue. Solving the fluidics was thus one of the important items that enabled the demonstration. While positive pressure on the fluid bags was important, air bubbles had to be eliminated inside the bags to allow sheath-flow operation. The copper shielding served to protect against external electromagnetic waves (EM) and possible interference from the device's electronics with the space station's communications systems. Small sample volumes are important for long-duration space travel and this was enabled by a microvolume sample loader that controlled the loading profiles. Furthermore, the expected technical impact seems to be limited, especially considering the availability of portable cytometry products with similar or even superior performance already in the market, like CyFlow Cube6 from Sysmex and Moxi Go II from Orflo. The technical impact (i.e. ability to measure cytometry-related parameters) is as that of a conventional cytometer; however, the device can be operated in spaceflight. It thus broadens the utility of the technique to space. As a point of interest, the CyFlow was developed as part of an international effort to send a cytometer to space (<https://www.sysmex-partec.com/company/who-is-sysmex-partec/history.html>) and Shapiro H, *Practical Flow Cytometry*. For the challenge of spaceflight reasons, there is no record of the CyFlow being launched, underscoring the exceptional challenges in sending one of these devices to operate successfully in space. One of the Orflo cytometers was tested onboard parabolic flights, but their cytometers are not standard cytometers since they lack side and forward scatter (substituted with impedance and forward extinction). The Orflo cytometers are thus not what would be preferred to obtain standard cytometry data. There is also no published data on the zero gravity experiments to suggest it would work in space. For these reasons, it seems reasonable for this paper to be published in a space-related specific journal. My additional comments are as follows:

(1) In space, electromagnetic wave distribution can vary significantly depending on time and location. For this reason, long-term signal monitoring in units of days, rather than seconds, seems necessary. Yes, the EM distribution is different and NASA's primary interest is to measure biological and human response to these different and summative conditions at defined time points. The idea is for astronauts to sample their blood and bodily fluids (i.e. weekly, daily) and when medical issues arise. Biological systems can also be studied as well in this manner. Small sample volumes enables the ability to analyze a larger number of time points since they are easier to obtain. The reviewer seems to suggest a type of in a loop experiment where samples (ideally the same sample) can be continuously monitored. This would be interesting, but was not the focus of NASA's Human Research Program.

(2) When experiments are repeated, the data presentation should include the standard deviation to provide an intuitive understanding of the degree of deviation. For example, in Figure 7, the authors should add a graph showing the deviation for each test result within each bead population.

We added SD, CV, mean tables in the supplement for all the runs. This should give the reader a sense of the deviation for all the runs.

(3) Figure 7 shows that the sample was degraded during the experiment. Is this damage attributed to the storage environment in space? Please provide further discussion on this issue. We added additional discussion in the text of the document. We do not know the relative contributions of the various factors that resulted in the degradation of the sample. The possibilities include room temperature storage of the sample or additional space radiation. The samples were adsorbed onto the surface of the beads and based on the data, it looks like the antibodies desorbed from one population to the other. Both factors could have contributed to the degraded sample.

(4) The authors should provide a representative plot of SSC vs FSC for cytometry data in Figure 7.

We included this as an additional panel. This is now Figure 8.

(5) Clinical tests seem to be limited by NASA regulations, but testing should be further conducted on clinically valuable samples.

Yes, that is the plan. NASA requires double containment of any biological samples. For the purpose of this demonstration, the goal was to demonstrate a functional technology in space.

Reviewer #2 (Remarks to the Author):

The authors present a compact flow cytometer with an extremely small footprint and whose performance matches with those of much larger, desktop commercial cytometers. The miniature cytometry-based analyzer, the rHEALTH ONE, is based on sheath-flow, uses sample volumes as small as 10uL and collects data on five separate channels simultaneously using a 10us bin interval. The FSC, SSC and the fluorescent signals compare well with those obtained using a standard cytometer thereby demonstrating the precision and accuracy of the measurements. The device was tested in zero-gravity conditions and performed well. These are excellent results and mark an important step in cytometry design.

Having said that the manuscript lacks design details, which makes reproducing their results impossible. These are good comments from the Reviewer #2 and we added additional detail for the reader. Keep in mind, that this was a modified commercially-available rHEALTH ONE in order to qualify for NASA's Commercial Off-The-Shelf (COTS) pathway to the ISS. The device, as designed, is the base unit and this was designed with high-level requirements from NASA. For payload development, additional ruggedization and modification was required and this included the zero gravity fluid vials and shielding. For example, it is not clear what design/scientific innovations enabled miniaturization of the optical unit. More details added in the paper as Figure 2 and the online methods. What was the design for the sheath-flow system, especially the flow cell where the laser beam strikes the particles? We included detailed images of the flow cell and the sheath flow area. What optics were used to shape the beam - was the cross-section of the laser beam, flat-hat or elliptical? We describe the dimensions of the elliptical beam shape. What lasers were used in the set-up - were these bought off the shelf? We provided the manufacturers of these off the shelf laser diodes. What was the power rating and stability of the lasers? We included this information in the device specification Table S1. How were the lasers cooled since it is well known that laser heating leads to instability of the beam intensity? We describe a fan-based cooling system. What type of photodiodes were used - were these bought off the shelf? We describe the vendor of the SiPMs. Commercial flow cytometers use photomultipliers due to the low intensity of fluorescence. Are these photomultipliers or avalanche photodiodes or standard photodiodes? We describe silicon photomultipliers from Hamamatsu. There are no details of the sensor circuit - what opamps were used? We provide our photon counting circuit for the reader. What was the amplification? A gain of 10 is described. What was the signal to noise ratio? Digital photon counting outputs were fed into the microcontroller. What type of microcontroller was used to capture signals? We describe the microcontroller. What design aspects enabled the device to withstand high-g forces? We provide further images on the optical module and device. Without these details it is not possible to assess the scientific/design innovation that went into building the compact design. Again, these details are needed to reproduce the results reported in the manuscript, without which the manuscript is not suitable for publication in a scientific journal. We included these details. As an example of a scientific article on design of cytometer, I suggest reading the paper by Habbersett et al (Cytometry Part A 71A: 809-817, 2007) who give full details of their cytometer including all components and circuits. This paper was helpful.

My second major criticism is regarding their FSC results. We looked into this more and would agree with the Reviewer #2. We reference Shapiro's *Practical Flow Cytometry*, 4th edition p275 as well to confirm the Reviewer #2 comments. They state that the FSC signal strength was found to be proportional to the particle diameter. We took this out. Note that the manufacturer (Spherotech) of the QC particles who show a mostly linear response on FSC with their QC process on their cytometer.

Data showing FSC measurement is proportional to the diameter of the beads, Cat. No. PPS-6K

The Gallios analysis (Figure S9 in our paper) show an increase in signal closer to the square relationships, but since it is four sizes, the true relationship could not be determined. Also based on Shapiro, cytometers differ in their ability to analyze the same particles.

This result is not correct. Mie theory predicts the FSC intensity to be proportional to the square of the particle size (or equivalently, particle cross-sectional area) - see "Absorption and Scattering of Light by Small Particles", Bohren & Huffman, Page 404 and Fig 13.10. These predictions have been confirmed via experiments (see for example, Pinnick and Auvermann, "RESPONSE CHARACTERISTICS OF KNOLLENBERG LIGHT-SCATTERING AEROSOL COUNTERS", US Army Electronics Research and Development Command, 1979). We based our comments on our observations of our cytometer and the benchmark Gallios cytometer. In both cases, we get an increasing FSC signal with increasing particle size. To Reviewer #2's point, we were more careful in stating a "proportional" response, but rather an "increasing" response. Only four sizes were examined so a square of the particle diameter relationship could not be determined. It should be noted that the FSC performance on various particle sizes is dependent on the specific instrument. Some instruments do not have a FSC channel that increases in intensity with diameter size (see Shapiro).

Based on the above observations, I am not inclined to recommend the manuscript for publication in Nature communications.

REVIEWERS' COMMENTS

Reviewer #1 (Remarks to the Author):

I have no further comments and still believe that the paper is better suited for a more specific journal focused on space-related topics.

Reviewer #2 (Remarks to the Author):

I have gone through the detailed response of the authors to the many queries raised by my earlier review. I have also read the revised manuscript. While I remain impressed with the results, contrary to my earlier understanding, the authors have clarified that the device in question was modified from an existing cytometer for space flight. In my opinion, the improvements over the existing setup namely, design of copper shielding, elimination of bubbles in fluid bags, and the design of the microvolume loader are not sufficiently novel to warrant publication in Nature communications. The scientific novelty is limited as it does not introduce substantially new insights or discoveries.